# (Sparse) Attention to the Details: Preserving Spectral Fidelity in ML-based Weather Forecasting Models

## Abstract

We introduce Mosaic, a probabilistic weather forecasting model that addresses two sources of spectral degradation in ML-based weather prediction: (1) training to predict the ensemble mean deterministically and (2) compressive encoding creating an information bottleneck. Mosaic combines learned functional perturbations for ensemble forecasting with block-sparse attention, a hardware-aligned formulation that shares keys and values across spatially adjacent queries, enabling each block to dynamically attend to the most relevant regions. By capturing arbitrarily long-range dependencies at linear cost, Mosaic processes high-resolution weather data without compression. Mosaic at $1.5°$ resolution matches or outperforms models trained on $0.25°$ data and achieves state-of-the-art results among $1.5°$ models on key upper-air variables, with individual ensemble members exhibiting near-perfect spectral alignment across all resolved frequencies.

## 1 Introduction

Accurate weather forecasts save lives and enable timely decisions during extreme events. Phenomena such as frontal zones and tropical cyclones cause catastrophic damage and span relatively short distances, requiring models that faithfully resolve fine spatial scales. Numerical weather prediction (NWP) systems achieve this by integrating the equations of fluid dynamics, thermodynamics, and radiative transfer, but at a computational cost that scales cubically with grid resolution.

ML-based weather prediction models (MLWPs; Bi et al. 2023; Lam et al. 2023; Bodnar et al. 2025) have emerged as efficient surrogates, generating 10-day forecasts in under 60 seconds on a single GPU, achieving a 1000-10000× speedup over NWP (Buizza et al., 2018; Bauer et al., 2020). However, these models struggle to reproduce the power spectra of atmospheric variables, systematically suppressing spectral power at fine scales (Bonavita, 2024). This happens for two main reasons. First, most MLWPs are deterministic and are trained to approximate the conditional mean over future states, which is inherently smoother than any individual realization. Probabilistic MLWPs (Price et al., 2023; Lang et al., 2024; Alet et al., 2025) address this limitation by producing ensemble members rather than a single mean prediction. Each member represents a plausible realization that can exhibit the sharp, fine-scale structures present in nature with spectral properties substantially closer to ground truth than their deterministic counterparts.

Second, most MLWPs use compressive encoding in their model architecture: they compress high-resolution weather data to a coarser mesh before processing, with a coarsening factor that typically far exceeds the increase in the number of feature channels, creating an information bottleneck shown to cause irreversible information loss in both vision (Wang et al., 2025) and weather models (Nguyen et al., 2023). Although compression-based approaches are efficient, since the bulk of computation occurs on the coarse mesh, they increase the complexity of the function that must be learned: fine-scale spatial variations that were originally distributed across neighboring mesh nodes become entangled within the channel dimensions of coarse nodes, requiring the model to jointly recover geometric structure and spectral content.

In this work, we introduce Mosaic: a probabilistic weather forecasting model designed to address both sources of spectral degradation. First, we follow Alet et al. (2025) and use learned functional perturbations to incorporate uncertainty, producing ensemble forecasts whose individ-

ual members preserve realistic spectral variability. Second, we propose block-sparse attention: a hardware-aligned formulation of native sparse attention (Yuan et al., 2025) that exploits the intrinsic locality of physical data by sharing keys and values across spatially adjacent queries, computing block-to-block interactions. Each block dynamically selects which regions of the grid to attend to, capturing arbitrarily long-range dependencies at linear cost. To ensure efficient memory access, we process data on the HEALPix mesh (Gorski et al., 1999), which offers contiguous storage and therefore allows us to operate on blocks. Together, these design choices make MOSAIC directly applicable to high-resolution grids without compression.

The main contributions of this work are:

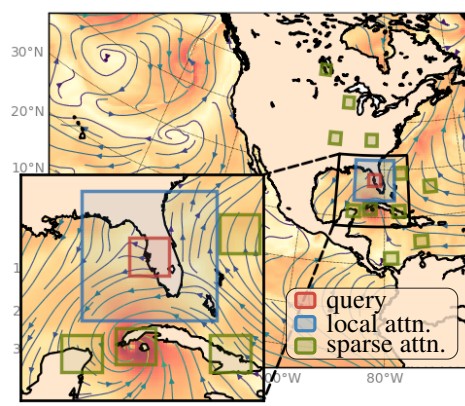

- We propose block-sparse attention, a hardware-aligned sparse attention mechanism that processes high-resolution grids at linear cost at native scale without compressive encoding.

- We introduce MOSAIC, a probabilistic weather forecasting model that combines block-sparse attention with learned functional perturbations, addressing both the architectural and statistical sources of spectral degradation in MLWPs.

- We demonstrate that MOSAIC, operating at 1.5° spatial resolution ($\approx$ 166km at the equator) without compression, outperforms the majority of state-of-the-art baselines trained on data with 6× higher spatial resolution and achieves state-of-the-art performance among models at the same 1.5° resolution on key upper-air variables, with ensemble members exhibiting near-perfect spectral alignment across all resolved frequencies.

Figure 1: Block-sparse attention for weather forecasting. Spatially close query tokens (red block over Tampa Bay) collectively attend to both local key-value pairs (blue block over Florida) and dynamically selected, spatially distributed ones (green blocks). Sparse attention enables capturing long-range dependencies in high-resolution weather data, critical for extreme events such as hurricane formation (note the eye in the inset).

## 2 RELATED WORK

### 2.1 EFFECTIVE RESOLUTION IN WEATHER FORECASTING

Abdalla et al. (2013) define the effective resolution of a weather prediction model as the smallest spatial scale it fully resolves – the wavelength at which the model's power spectrum starts to deviate from the ground truth. Multiple studies (Bonavita, 2024; Husain et al., 2025) show that MLWPs fail to reproduce realistic spectra and therefore exhibit low effective resolution, struggling to resolve sharp phenomena such as frontal zones and tropical cyclones that span 50-80 km, despite being trained on data at 28 km spatial resolution. Gupta et al. (2025) find systematic underestimation of spectral power at mesoscales (10-100 km) across multiple MLWPs. Li et al. (2025) demonstrate that Pangu (Bi et al., 2023) underestimates kinetic energy at wavelengths below 1000 km and fails to replicate the characteristic $-\frac{5}{3}$ spectral slope of physics-based models.

Several strategies address this spectral degradation. Subich et al. (2025) modify the training objective to penalize spectral discrepancy directly, improving GraphCast's effective resolution from 1,250 to 160 km. Similarly, Bonev et al. (2025) also optimize in the spectral domain, achieving realistic spectra at subseasonal lead times. Hybrid approaches (Kochkov et al., 2023; Husain et al., 2025) combine ML with physics-based solvers, leveraging the numerical backbone for fine-scale structure. Post-hoc methods (Lippe et al., 2023; Oommen et al., 2024) condition diffusion models on smooth predictions to recover high-frequency content. Most directly related to our work, Baño-Medina et al. (2025); Nordhagen et al. (2025) process on the original high-resolution mesh with message-passing neural networks and observe significantly better spectral correspondence. We follow the same principle, but replace the fixed graphs with sparse attention, which dynamically determines interactions based on the current weather state.

## 2.2 COMPRESSION EFFECT ON EXPRESSIVITY

The effect of compressive encoding on model expressivity is well studied in computer vision, where patchification, the core tokenization strategy of vision transformers (Dosovitskiy et al., 2020), reduces computational cost by compressing spatial information. Wang et al. (2025) demonstrate irreversible information loss caused by patchification and show that test loss declines consistently as patch size decreases, reaching optimal performance at $1 \times 1$ patches. Moreover, reducing patch size is more beneficial than increasing model parameters, indicating that added capacity cannot compensate for compression-induced information loss. The pattern holds in weather forecasting, where Nguyen et al. (2023) show that decreasing patch size improves forecast accuracy.

For atmospheric data, where fine-scale spatial structure is associated with extreme weather events, this information loss is especially consequential. We avoid compressive encoding entirely: rather than projecting high-resolution data onto a coarse mesh, we operate directly at native resolution via block-sparse attention to preserve details.

## 2.3 ULTRA-SCALE GRID PROCESSING

Avoiding compression shifts the bottleneck to efficiently processing the original ultra-scale data. Standard attention (Vaswani et al., 2017) scales quadratically with sequence length; FlashAttention (Dao et al., 2022) reduces the memory cost to linear, but still has the quadratic complexity. Linear attention (Yang et al., 2024; 2023) achieves linear cost by replacing the softmax with a kernel-based approximation that maintains a fixed-size state, but sacrifices the input-dependent selectivity that makes standard attention expressive. Native Sparse Attention (NSA; Yuan et al. 2025) resolves this trade-off by computing the dot product over a dynamically selected subset of key-value pairs per query, retaining softmax expressivity at linear cost.

In scientific computing, the need to process large-scale physical simulations has led to complementary approaches. Holzschuh et al. (2025) handle ultra-scale grids ($256^3$) by partitioning the domain into non-overlapping subdomains processed in parallel. Alkin et al. (2024); Wu et al. (2024) use learnable pooling to compress the input into a coarser representation where the bulk of computation occurs. Zhdanov et al. (2025); Brita et al. (2025) avoid pooling and instead use hierarchical trees to impose structure on irregular data, enabling sparse attention with linear cost.

We adapt sparse attention for high-resolution weather grids but further exploit spatial locality: rather than selecting key-value blocks independently per query, we group spatially proximate queries into blocks that jointly select which regions to attend to. This block-level sparsity aligns well with GPU memory access patterns, enabling efficient processing of grids with hundreds of thousands of points without domain decomposition or lossy pooling.

## 3 THEORETICAL BACKGROUND

### 3.1 PROBLEM FORMULATION

We frame weather forecasting as sampling from the conditional distribution $p\left(X^t \mid X^{t-1}, \ldots, X^{t-H}\right)$ over the next atmospheric state $X^t$ given the history of previous states, where each state consists of both surface- and pressure-level atmospheric variables on a latitude-longitude grid. A $T$-step ensemble forecast is generated by sampling autoregressively: $X^t \sim p\left(X^t \mid X^{t-1}, \ldots, X^{t-H}\right), \quad t = 1, \ldots, T$. Drawing multiple trajectories from this process yields an ensemble that quantifies forecast uncertainty.

### 3.2 HIERARCHICAL SPHERE TESSELLATION

HEALPix (Gorski et al., 1999) is a hierarchical, equal-area tessellation of a sphere into four-sided polygons (pixels). The sphere is partitioned into a tree-like hierarchy starting from 12 base pixels (4 around each pole, 4 around the equator), with each pixel recursively subdivided into four children, see Fig. 2. At a

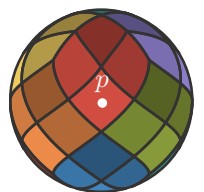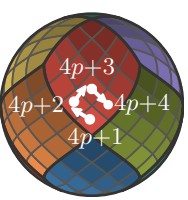

Figure 2: HEALPix mesh refinement.

given resolution, each pixel covers an identical surface area, ensuring that signal sampling and noise integration are geographically unbiased, unlike traditional latitude-longitude grids, which over-sample near the poles.

The size parameter $N_{\text{side}}$ defines a HEALPix mesh and determines both the total number of pixels $12N_{\text{side}}^2$ and the approximate angular resolution (see Table 4 in the appendix). The pixels are organized in memory such that spatially close regions occupy consecutive indices[1]. A pixel with index $p$ at resolution $N_{\text{side}}$ subdivides into four children with consecutive indices $4p, 4p+1, 4p+2, 4p+3$ at resolution $2N_{\text{side}}$. This contiguous, hierarchical layout makes HEALPix particularly suited for block-based computation: loading a spatially local block of pixels requires a single coalesced memory read rather than the scattered accesses needed on a latitude-longitude grid. Several ML-based weather models already exploit this structure (Ramavajjala, 2024; Linander et al., 2025; Karlbauer et al., 2023).

### 3.3 NATIVE SPARSE ATTENTION

Native Sparse Attention (NSA; Yuan et al. (2025)) addresses the quadratic complexity of standard self-attention by restricting each query's interactions to a dynamically determined subset of keys and values. The computation is organized into three branches – compression, selection and local – whose outputs $\mathbf{o}_i^{CG}$ (coarse-grained), $\mathbf{o}_i^{FG}$ (fine-grained) and $\mathbf{o}_i^L$ (local) are combined via gating:

$$\mathbf{o}_i = g^{CG}(\mathbf{x}_i) \cdot \mathbf{o}_i^{CG} + g^{FG}(\mathbf{x}_i) \cdot \mathbf{o}_i^{FG} + g^L(\mathbf{x}_i) \cdot \mathbf{o}_i^L \tag{1}$$

where $g^{CG}, g^{FG}, g^L$ are learnable linear gating functions.

**Compression** Let $\{B_1, \ldots, B_m\}$ be a partition of tokens into $m$ non-overlapping blocks. For each $B_j$, let $\mathbf{K}_j = \{\mathbf{k}_l \mid l \in B_j\}$ denote the set of keys in the block. A block representation is computed via a learnable function $\varphi$, $\bar{\mathbf{k}}_j = \varphi(\mathbf{K}_j)$. Coarse-grained values are obtained as $\bar{\mathbf{v}}_j = \varphi(\mathbf{V}_j)$ with $\mathbf{V}_j = \{\mathbf{v}_l \mid l \in B_j\}$. Coarse-grained attention between query $i$ and each block is then evaluated, with attention scores retained for the selection branch:

$$a_{ij} = \frac{\exp\left(\mathbf{q}_i^\top \bar{\mathbf{k}}_j / \sqrt{d_k}\right)}{\sum_{l=1}^m \exp\left(\mathbf{q}_i^\top \bar{\mathbf{k}}_l / \sqrt{d_k}\right)}, \quad \mathbf{o}_i^{CG} = \sum_{j=1}^m a_{ij} \bar{\mathbf{v}}_j. \tag{2}$$

The branch both captures global context and guides sparsification in the selection branch.

**Selection** For each query $i$, NSA selects the top-$n$ blocks with the highest coarse-grained attention scores, $\mathcal{S}_i = \text{top-}n\,(a_{i,:})$, and computes fine-grained attention over keys and values *within* the selected blocks:

$$\mathbf{o}_i^{FG} = \sum_{j \in \mathcal{S}_i} \sum_{l \in B_j} \frac{\exp\left(\mathbf{q}_i^\top \mathbf{k}_l / \sqrt{d_k}\right)}{Z_i} \mathbf{v}_l \tag{3}$$

where $Z_i = \sum_{j \in \mathcal{S}_i} \sum_{l \in B_j} \exp\left(\mathbf{q}_i^\top \mathbf{k}_l / \sqrt{d_k}\right)$ is the normalizing constant. By operating at full resolution, the selection branch preserves fine-scale detail while capturing long-range interactions.

**Local Attention** The local branch applies standard attention for each query $i$ over keys and values within a sliding window, yielding $\mathbf{o}_i^L$. This frees the compression and selection branches to focus on long-range interactions.

## 4 MOSAIC

Tobler's first law of geography states that "nearby things are more related than distant things" (Tobler, 1970). This principle motivates two important design choices in MOSAIC: (1) representing data on the HEALPix mesh, where spatially close points occupy contiguous memory and (2) grouping spatially adjacent tokens into blocks that jointly determine which regions of the globe to attend to.

---

[1]This corresponds to the NESTED ordering scheme. We refer the reader to (Gorski et al., 1999) for information about the alternative RING scheme.

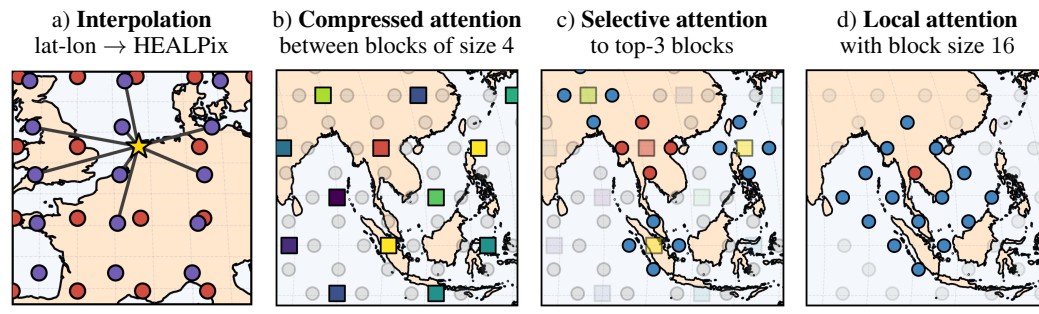

a) **Interpolation**
lat-lon → HEALPix

b) **Compressed attention**
between blocks of size 4

c) **Selective attention**
to top-3 blocks

d) **Local attention**
with block size 16

Figure 3: Block-sparse attention for weather forecasting. **(a)** Weather data is interpolated from a latitude-longitude grid (red) to the HEALPix mesh (purple) via cross-attention. **(b–d)** The three branches of block-sparse attention, illustrated for a single query block (red): **(b)** Compression computes attention between coarse-grained block representations (squares; color indicates attention score). **(c)** Selection attends at full resolution to the top-n key blocks. **(d)** Local attention captures fine-grained interactions within local blocks.

## 4.1 DATA REPRESENTATION ON HEALPIX

Weather forecast data is traditionally stored on latitude-longitude grids, where points are ordered row-wise along parallels. This layout places spatially close points far apart in index space, requiring scattered memory accesses to load a block of neighboring pixels. We instead operate on the HEALPix mesh, which guarantees that spatial neighbors occupy contiguous memory locations, enabling coalesced GPU reads and hardware-aligned block computation.

**Interpolating Between Grids** To transfer data from the latitude-longitude grid to the HEALPix mesh, we employ the cross-attention interpolation scheme of Wessels et al. (2025). For each target point $i$ on the HEALPix mesh and neighboring source point $j$ on the latitude-longitude grid, queries are computed from the relative position $\mathbf{p}_{ij}$, while keys and values are derived from source features:

$$\mathbf{q}_{ij} = W_q \frac{\mathbf{p}_{ij}}{\|\mathbf{p}_{ij}\|}, \quad \mathbf{k}_j = W_k \mathbf{x}_j, \quad \mathbf{v}_j = W_v \mathbf{x}_j, \quad \mathbf{o}_i = \sum_{j \in \mathcal{N}_i} \mathrm{softmax}_j \left( \mathbf{q}_{ij}^\top \mathbf{k}_j / \sqrt{d} \right) \mathbf{v}_j \quad (4)$$

where $\mathcal{N}_i$ denotes the set of source grid points neighboring $i$.

## 4.2 BLOCK-SPARSE ATTENTION

BSA retains the three-branch structure of NSA – compression, selection, and local – combined via learned gating (Eq. 1). The key difference is that sparsity operates at the block level: rather than each query token independently selecting which key blocks to attend to, tokens within a query block jointly select key blocks. This reduces compression from token-to-block to block-to-block interactions and amortizes the selection cost across all tokens in a block.

**Compression** We partition $N$ tokens into $m$ non-overlapping blocks $\{B_1, \ldots, B_m\}$ and compute coarse-grained representations of queries, keys, and values via mean pooling $\varphi$:

$$\bar{\mathbf{q}}_i = \varphi(\mathbf{Q}_i), \quad \bar{\mathbf{k}}_j = \varphi(\mathbf{K}_j), \quad \bar{\mathbf{v}}_j = \varphi(\mathbf{V}_j). \quad (5)$$

Attention is computed at the block level between query block $i$ and key block $j$, with the coarse-grained output then distributed to all tokens within each query block:

$$\bar{a}_{ij} = \frac{\exp\left(\bar{\mathbf{q}}_i^\top \bar{\mathbf{k}}_j / \sqrt{d_k}\right)}{\sum_{l=1}^m \exp\left(\bar{\mathbf{q}}_i^\top \bar{\mathbf{k}}_l / \sqrt{d_k}\right)}, \quad \bar{\mathbf{o}}_i^{CG} = \sum_{j=1}^m \bar{a}_{ij} \bar{\mathbf{v}}_j, \quad \mathbf{o}_l^{CG} = \bar{\mathbf{o}}_i^{CG} \quad \forall l \in B_i. \quad (6)$$

Coarse-grained attention captures broad spatial patterns in a computationally feasible manner. For instance, a block over the Netherlands might attend strongly to blocks over the North Atlantic or the Arctic, identifying synoptic-scale influences before the selection branch zooms in on details.

**Selection**    For each query block $i$, the top-$n$ key blocks with highest coarse-grained attention scores are selected, $\bar{\mathcal{S}}_i = \text{top-}n\left(\bar{a}_{i,:}\right)$. Fine-grained attention is then computed between all tokens in query block $i$ and all tokens in the selected key blocks, capturing long-range dependencies:

$$\mathbf{o}_l^{FG} = \sum_{j \in \bar{\mathcal{S}}_i} \sum_{t \in B_j} \frac{\exp\left(\mathbf{q}_l^\top \mathbf{k}_t / \sqrt{d_k}\right)}{Z_l} \mathbf{v}_t \quad \forall l \in B_i \tag{7}$$

where $Z_l = \sum_{j \in \bar{\mathcal{S}}_i} \sum_{t \in B_j} \exp\left(\mathbf{q}_l^\top \mathbf{k}_t / \sqrt{d_k}\right)$ is the normalizing constant. The selection branch resolves fine-scale structure within the chosen regions, for example, by capturing how a specific frontal zone over the Atlantic influences local conditions in Western Europe.

**Local Attention**    To capture fine-grained local interactions, we compute attention within large blocks of pixels independently. Unlike NSA's sliding window, which would require handling irregular boundaries on the sphere, block attention aligns naturally with the HEALPix structure and can be computed in parallel. The local branch resolves fine-grained structure within each region, such as temperature gradients across a coastline or wind patterns within a valley, freeing other branches to focus on long-range interactions.

We implement BSA in Triton (Tillet et al., 2019) following the approach of FlashAttention (Dao et al., 2022); see Appendix A.6 for details on implementation and computational cost.

### 4.3    MODEL ARCHITECTURE

Weather dynamics spans a wide range of spatial scales, from planetary waves spanning thousands of kilometers to mesoscale convective systems tens of kilometers across. To capture this multi-scale structure, we adopt a U-Net architecture that processes data at progressively coarser resolutions in the encoder path, then refines predictions back to the original resolution in the decoder path. Each resolution level consists of block-sparse attention layers that capture interactions at that scale. Skip connections between corresponding encoder and decoder levels preserve fine-grained information that might otherwise be lost during coarsening.

**Coarsening**    The encoder path progressively coarsens the HEALPix mesh. At each coarsening step, a parent pixel at resolution $N_{\text{side}}$ aggregates features from its four children at resolution $2N_{\text{side}}$ via learnable pooling: $\mathbf{x}_{\text{parent}} = W_x^\downarrow \mathbf{X}_c + W_p^\downarrow \Delta \mathbf{P}_c$, where $\mathbf{X}_c = [\mathbf{x}_{c_i}]_{i=1}^4$ and $\Delta \mathbf{P}_c = [\Delta \mathbf{p}_{c_i}]_{i=1}^4$ are the stacked child features and relative positions, with $\Delta \mathbf{p}_{c_i} = \mathbf{p}_{c_i} - \mathbf{p}_{\text{parent}}$. Here $W_x^\downarrow, W_p^\downarrow$ are learnable projections. At the coarsest resolution, a bottleneck stage applies additional transformer blocks before the decoder path begins.

**Refinement**    The decoder path progressively refines features back to the original resolution. At each refinement step, a parent pixel at resolution $N_{\text{side}}$ predicts features for its four children at resolution $2N_{\text{side}}$: $\mathbf{X}_c = W_x^\uparrow \mathbf{x}_{\text{parent}} + W_p^\uparrow \Delta \mathbf{P}_c$. The corresponding encoder features are then added to the refined features via skip connections.

**Overall architecture**    Input features that include both dynamic (e.g. 2-meter temperature, 10-meter wind components) and static variables (e.g. land-sea mask, soil type) are combined with sinusoidal time (day and year) encodings (Bodnar et al., 2025) and projected to the hidden dimension via a two-layer MLP followed by interpolation to the HEALPix mesh (see Eq. 4).

The data then passes through the U-Net, where a sequence of transformer blocks is applied at each resolution level. Each block has a standard pre-norm structure (Touvron et al., 2023) with block-sparse attention and SwiGLU (Shazeer, 2020) feed-forward network.

$$\mathbf{x} \leftarrow \mathbf{x} + \text{Attention}(\text{RMSNorm}(\mathbf{x})); \qquad \mathbf{x} \leftarrow \mathbf{x} + \text{FFN}(\text{RMSNorm}(\mathbf{x}), \mathbf{z}) \tag{8}$$

We use grouped query attention (GQA; Ainslie et al. (2023)) to reduce memory bandwidth requirements, with queries attending to a number of shared key-value heads. We apply 2D rotary positional embeddings (RoPE; Heo et al. (2024)) to queries and keys, encoding longitude and latitude positions. The head dimension is split equally between the two coordinates, with each half receiving standard RoPE encoding for its respective angle in radians. After the decoder, features are interpolated back to the latitude-longitude grid and projected to the output dynamic variables to predict the next weather state.

### 4.4 UNCERTAINTY QUANTIFICATION

Block-sparse attention addresses the architectural source of spectral degradation by avoiding compression. To address the statistical source, MOSAIC produces ensemble forecasts via learned functional perturbations (Alet et al., 2025), a form of Bayesian neural networks (Blundell et al., 2015). The key idea is to inject noise into the parameters of the model, resulting in a single weight perturbation affecting the entire forecast in a globally consistent way. Alet et al. (2025) condition parameters of layer normalization layers on the noise vector; we instead inject noise into SwiGLU layers as bias added inside the gate:

$$\text{cSwiGLU}(x, z) = \left( \sigma\left(xW_g + z\right) \odot \left(xW_v\right) \right) W_{out} \tag{9}$$

where $\odot$ denotes element-wise multiplication, and $\sigma(x) = \frac{x}{1+e^{-x}}$ is the Swish activation. We found simple additive noise injection in the SwiGLU gate to work best. Mathematically, this induces an effective output projection $W_{out}^{\text{eff}} = \text{diag}(s) \cdot W_{out}$, where scaling factors $s_i$ are drawn from input-dependent distributions determined by the gate activations $xW_g$. The learned weights $W_{out}$ define the mean structure, while noise $z$ flowing through the input-dependent nonlinearity defines a learned, adaptive covariance. This produces structured uncertainty as the model learns which features should vary across ensemble members based on the input and a small number of latent variables.

**Training with CRPS** The probabilistic model is trained by optimizing the Continuous Ranked Probability Score (CRPS; Alet et al. (2025)), a proper scoring rule for univariate distributions that encourages both accurate and well-calibrated probabilistic forecasts. We use an unbiased CRPS estimator (Zamo & Naveau, 2018), which is given by

$$\text{CRPS}(x^{1:N}, y) := \frac{1}{N} \sum_n |x^n - y| - \frac{1}{2N(N-1)} \sum_{n,n'} |x^n - x^{n'}|. \tag{10}$$

for a single variable $y \in \mathbb{R}$ and an ensemble $x^{1:N}$. The CRPS decomposes into a reliability term penalizing deviation from the ground truth and a sharpness term encouraging ensemble spread only where warranted by true uncertainty. The training loss is $\mathcal{L} := \frac{1}{|D|} \sum_d \frac{1}{G} \sum_i \alpha_i \text{CRPS}(x_{i,d}^{1:N}, y_{i,d})$, where $d$ indexes elements of the dataset batch $D$, $i$ indexes $G$ variable-level tuples, and $\alpha_i$ is the corresponding loss weight taken from Lam et al. (2023).

## 5 EXPERIMENTS

We aim to answer the following research questions:

- **RQ1:** How does MOSAIC compare against SOTA MLWPs in terms of forecasting skill?
- **RQ2:** Does MOSAIC preserve spectral fidelity across all resolved frequencies?
- **RQ3:** Are MOSAIC's ensemble forecasts well-calibrated?

**Data and training** We pretrain on ERA5 reanalysis (1979–2018, validation 2019). Finetuning and evaluation depend on the experiment. For the comparison against state-of-the-art 1.5° models, we further finetune on ERA5 (2006–2018) (Couairon et al., 2024) and evaluate on 2020. For the comparison against state-of-the-art 0.25° models, we finetune on HRES-fc0 analysis (2016–2021) and evaluate on 2022. In both cases, finetuning includes progressively longer autoregressive rollouts, see Appendix A.4 for details. All data is at 1.5° resolution (121×240 grid). MOSAIC contains 214M parameters and is trained with the Muon optimizer (Jordan et al., 2024), to our knowledge, the first application of Muon to MLWPs. Training takes 2 days on 8× H100 GPUs in float16 (Table 2); full hyperparameters are in Appendix A.4.

**Baselines** We compare against state-of-the-art medium-range MLWPs, both deterministic (Graph-Cast (Lam et al., 2023), Aurora (Bodnar et al., 2025), Pangu-Weather (Bi et al., 2023)) and probabilistic (FGN (Alet et al., 2025), GenCast (Price et al., 2023)). For the 1.5° comparison (Table 1), we additionally compare against models trained at comparable resolution: ArchesWeather and ArchesWeatherGen (Couairon et al., 2024), Stormer (Nguyen et al., 2024), SphericalCNN (Esteves et al., 2023), and NeuralGCM (Kochkov et al., 2023). We also report the performance of NWP systems (IFS-ENS, IFS-HRES (ECMWF)). Training resources of baselines are given in Table 2.

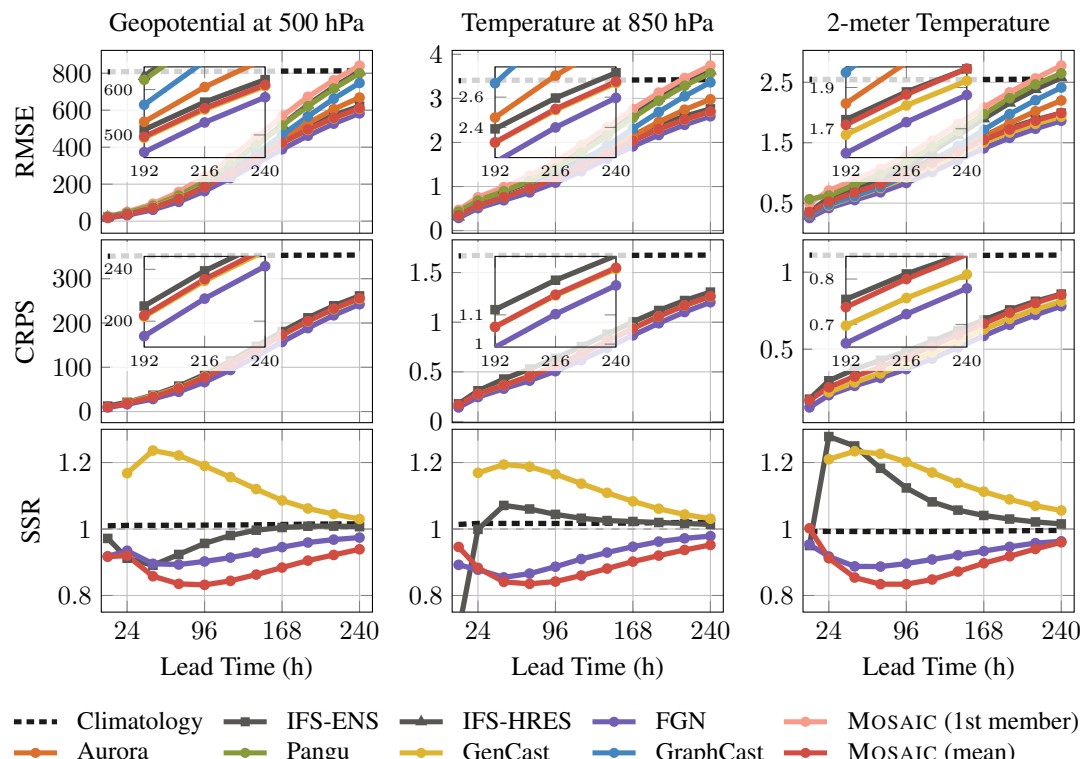

Figure 4: Forecast skill evaluated against IFS HRES Analysis, 2022 test year. All forecasts regridded to 1.5° resolution. Baselines operate at 0.25°, while MOSAIC operates at 1.5°.

**Evaluation protocol** We generate forecasts with an ensemble size of 48 for evaluation. Following the WeatherBench 2 protocol (Rasp et al., 2023), we initialize forecasts every day of a test year at 00:00 and 12:00 UTC; all forecasts and ground truths are first-order conservatively regridded to 1.5° before computing metrics. To evaluate forecasting skill, we report the latitude-weighted root mean square error (RMSE), as well as the fair CRPS (Rasp et al., 2023) of the ensemble (**RQ1**). Metrics are computed at each grid point, then globally area-weighted and averaged for each variable and pressure level separately. To evaluate spectral fidelity, we report spectra of global kinetic energy (**RQ2**). To evaluate ensemble reliability, we report the spread-to-skill ratio (**RQ3**).

RQ1: FORECASTING SKILL COMPARISON

Fig. 4 shows RMSE and CRPS results across three headline variables: geopotential at 500 hPa (Z500), temperature at 850 hPa (T850), and 2-meter temperature (T2m). At 10-day lead time, MO-SAIC achieves strong performance in terms of RMSE: on Z500 and T850, MOSAIC outperforms IFS-ENS, Aurora, Pangu, and GraphCast, while matching GenCast; on T2m, MOSAIC matches IFS-ENS while outperforming Aurora, Pangu, and GraphCast. Furthermore, MOSAIC demonstrates competitive CRPS scores. At 240 h, MOSAIC achieves Z500 CRPS of 256, close to GenCast (254) and below IFS-ENS (261). On T850, MOSAIC matches GenCast and outperforms IFS-ENS. On T2m, MOSAIC (0.85) matches IFS-ENS (0.86) and approaches GenCast (0.81).

Overall, our answer to **RQ1** is that MOSAIC's block-sparse attention operating at a lower native resolution without compression can match or exceed the performance of SOTA models, including GenCast. Tables 9 and 10 provide RMSE at 24 h and 240 h for a broader set of headline variables; RMSE, CRPS, and spread-to-skill curves for the remaining variables (U10, V10, MSL, U850, V850, Q700) are shown in Appendix B.1, along with qualitative unrolling trajectories.

**Comparison with 1.5° resolution models** Table 1 reports RMSE for headline variables (remaining variables in Table 8). MOSAIC achieves the best Z500 RMSE among all ≥1° models, outperforming ArchesWeather-Mx4 (41.94) and ArchesWeatherGen (41.92). On wind components, MOSAIC also leads: U850 of 1.121 versus 1.163 for ArchesWeatherGen, and V850 of 1.146 versus

|  | RES. | STEP | Z500 | T850 | Q700 | U850 | T2M | SP |
|---|---|---|---|---|---|---|---|---|
| IFS HRES | 0.1° | 1H | 42.30 | 0.625 | 556.0 | 1.186 | 0.513 | 60.16 |
| SPHERICALCNN | 1.4° | 6H | 54.43 | 0.738 | 591.0 | 1.439 | N/A | N/A |
| STORMER | 1.4° | 24H | 45.21 | 0.607 | 527.9 | 1.165 | 0.664 | 55.39 |
| NEURALGCM ENS (50) | 1.4° | 12H | 43.99 | 0.658 | 542.5 | 1.239 | N/A | N/A |
| ARCHESWEATHER-Mx4 | 1.5° | 24H | 41.94 | **0.593** | **513.3** | 1.172 | **0.517** | 52.23 |
| ARCHESWEATHERGEN (MEAN) | 1.5° | 24H | 41.92 | 0.595 | 518.4 | 1.163 | 0.541 | **51.71** |
| MOSAIC (MEAN; OURS) | 1.5° | 6H | **40.80** | 0.611 | 530.7 | 1.121 | 0.705 | 54.74 |
| MOSAIC$_\Delta$ (MEAN; OURS) | 1.5° | 6H | 41.00 | 0.600 | 526.1 | **1.119** | 0.677 | 54.77 |

Table 1: Comparison of coarse resolution MLWPs on RMSE scores for key weather variables with 24h lead-time against ERA5 data, 2020 test year. Best scores in **bold**, second best underlined. Results for additional variables (V850, U10m, V10m) are provided in Table 8.

1.190 (Table 8). On surface variables (T2m, SP, U10m, V10m), ArchesWeather variants perform better, likely benefiting from their 24 h prediction step versus MOSAIC's 6 h step, which accumulates errors over four autoregressive rollouts to reach 24 h. We also report MOSAIC$_\Delta$, a variant that predicts the state increment $\Delta X^t = X^t - X^{t-1}$ rather than the full state.

RQ2: SPECTRAL FIDELITY PRESERVATION

Due to compute constraints, MOSAIC operates at 1.5° resolution, which limits our spectral analysis to the frequencies resolved at this grid spacing, in contrast to the 0.25° spectra shown in Fig. 12 for SOTA models. Within this resolution, however, Fig. 5 demonstrates strong spectral alignment: the first ensemble member is coherent with HRES-fc0 across all resolved frequencies, with no systematic suppression of high-frequency power (see also Fig. 10, 11). The ensemble mean exhibits expected deviation at higher frequencies due to smoothing from averaging. This result validates our core hypothesis: by avoiding spatial compression and oper-

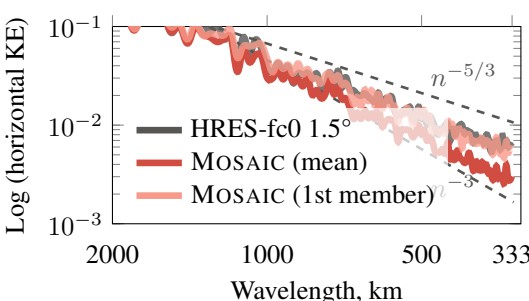

Figure 5: Global kinetic energy spectra at 10-meter height for 10-day MOSAIC forecasts compared to HRES-fc0 1.5° ground truth.

ating on native-resolution grids, MOSAIC preserves spectral characteristics up to the grid's Nyquist limit. While we cannot extrapolate how this would translate to finer resolutions, the absence of architectural spectral degradation at 1.5° is encouraging for the future work at 0.25° resolution.

RQ3: ENSEMBLE CALIBRATION

The bottom row of Fig. 4 shows the spread-to-skill ratio across lead times. Well-calibrated ensembles should exhibit ratios close to 1, with values below 1 indicating overconfidence and values above 1 indicating underconfidence. MOSAIC achieves ratios ranging from 0.83 at medium lead times to 0.95 at 240h, closely matching FGN (0.89–0.97), while GenCast shows persistent underconfidence (1.03–1.24). This demonstrates that noise injection in SwiGLU gates produces well-calibrated uncertainty estimates on par with state-of-the-art probabilistic MLWPs.

## 6  CONCLUSION

We introduced MOSAIC, a probabilistic weather forecasting model that addresses spectral degradation in MLWPs via (1) block-sparse attention, which processes high-resolution grids without compression, and (2) learned functional perturbations, which produce ensemble members with realistic spectral variability. Despite operating at 6× coarser resolution than most baselines, MOSAIC matches or exceeds state-of-the-art deterministic and ensemble systems while requiring 30× less training compute. When compared against models at the same 1.5° resolution, MOSAIC achieves state-of-the-art performance on key upper-air variables. We leave scaling MOSAIC to 0.25° grids and extending to subseasonal timescales for future work.

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

## A    IMPLEMENTATION DETAILS

### A.1    COMPUTATIONAL RESOURCES

Table 2 compares MOSAIC with baseline MLWPs. All experiments were conducted on $8\times$ NVIDIA H100 GPUs. Training requires 2 days (16 GPU-days total) using float16 precision throughout.

Table 2: Comparison of MOSAIC with baseline MLWPs. MOSAIC achieves competitive performance despite operating at coarser resolution with significantly reduced computational resources.

| Model | Params | Precision | Resolution | Training | Duration |
|---|---|---|---|---|---|
| GraphCast | 36M | fp32 | 0.25° | 32 TPUv4 | 4 weeks |
| GenCast | 57.5M | fp32 | 0.25° | 32 TPUv5 | 5 days |
| FGN | 57.5M | fp32 | 0.25° | 490 TPUv5p/v6e | 3 days* |
| Pangu | 277M | fp32 | 0.25° | 192 V100 | 15 days |
| Aurora | 1,259M | bf16 | 0.25° | 32 A100 | 2.5 weeks |
| MOSAIC | 214M | fp16 | 1.5° | 8 H100 | 2 days |

*Total of 490 TPU-days compute

### A.2    DATA DESCRIPTION

Table 3 lists all input and output variables used by MOSAIC. The model takes as input 4 surface-level variables and 6 pressure-level variables at 13 pressure levels, yielding $4+6\times 13 = 82$ dynamic channels per timestep. Additionally, 3 static fields are concatenated along the feature axis together with Cartesian coordinates $(x, y, z)$ on the unit sphere and 4 sinusoidal time embeddings (sine and cosine of day-of-year and year progress).

Table 3: Complete list of variables used by MOSAIC. Pressure-level variables are provided at 13 levels: {50, 100, 150, 200, 250, 300, 400, 500, 600, 700, 850, 925, 1000} hPa.

| Variable | Type | Unit | Channels |
|---|---|---|---|
| *Surface-level variables (dynamic)* | | | |
| 2-meter temperature | Surface | K | 1 |
| 10-meter U-wind | Surface | m/s | 1 |
| 10-meter V-wind | Surface | m/s | 1 |
| Mean sea level pressure | Surface | Pa | 1 |
| *Pressure-level variables (dynamic)* | | | |
| Geopotential | Pressure | $m^2/s^2$ | 13 |
| Specific humidity | Pressure | kg/kg | 13 |
| Temperature | Pressure | K | 13 |
| U-component of wind | Pressure | m/s | 13 |
| V-component of wind | Pressure | m/s | 13 |
| Vertical velocity | Pressure | Pa/s | 13 |
| *Static variables* | | | |
| Surface geopotential | Static | $m^2/s^2$ | 1 |
| Land-sea mask | Static | 0–1 | 1 |
| Soil type | Static | categorical | 1 |
| **Total dynamic channels per timestep: 82** | | | |

**Data normalization.**    All dynamic variables are standardized per channel (i.e., per variable and pressure level) using the mean and standard deviation computed over all spatial positions and all training timesteps via Welford's online algorithm. Each input state is normalized as $\hat{x} = (x - \mu)/\sigma$, where $\mu$ and $\sigma$ are the per-channel training statistics. The same normalization is applied to target states: $\hat{y} = (y - \mu)/\sigma$. The model predicts normalized next states directly, and predictions are

denormalized at inference time via $x = \hat{x} \cdot \sigma + \mu$. Static variables are normalized independently using their own spatial mean and standard deviation and augmented with Cartesian coordinates $(x, y, z)$ on the unit sphere. During pretraining, normalization statistics are computed on the ERA5 training split (1979–2018). During finetuning, they are recomputed on the HRES-fc0 analysis data (2016–2021).

**Data splits.** During pretraining on ERA5 reanalysis, we use years 1979–2018 for training, 2019 for validation, and 2020 for testing. During finetuning, we train on HRES-fc0 analysis data from 2016–2021 without a held-out validation set and evaluate on the year 2022. All data is sampled at 6-hourly temporal resolution.

**Data source and resolution.** All data is obtained from the WeatherBench 2 repository (Rasp et al., 2023) as Zarr archives on Google Cloud Storage. Both ERA5 and HRES datasets are conservatively remapped from their native grids to a $240 \times 121$ equiangular latitude-longitude grid (1.5° resolution) at 6-hourly intervals. No additional regridding or interpolation is applied to the data prior to model input.

Table 4: HEALPix mesh resolutions for different $N_{\text{side}}$ values.

| $N_{\text{side}}$ | 32 | 64 | 128 | 256 |
|---|---|---|---|---|
| # pixels | 12,288 | 49,152 | 196,608 | 786,432 |
| resolution | 1.83° | 0.92° | 0.46° | 0.23° |

## A.3 MODEL ARCHITECTURE

Table 5 lists the detailed architecture configuration. The U-Net comprises two coarse-graining stages and a bottleneck, with encoder and decoder layers at each stage.

Table 5: Model architecture details.

| Stage | nside | Dim | Heads | Enc. Depth | Dec. Depth | MLP Ratio |
|---|---|---|---|---|---|---|
| Stage 1 | 64 | 768 | 12 | 4 | 2 | 4.0 |
| Stage 2 | 32 | 1024 | 16 | 4 | 2 | 4.0 |
| Bottleneck | 16 | 1280 | 20 | 2 | | 4.0 |
| **Global parameters** | | | | | | |
| Total parameters: 214M | | | | | | |
| GQA ratio: 4, QKV compression ratio: 1 | | | | | | |
| RoPE: enabled ($\theta = 10000$), QK norm: disabled | | | | | | |
| RMSNorm elementwise affine: disabled | | | | | | |
| History steps: 4, Noise dim: 32, $k$-neighbors: 24 | | | | | | |
| **Block-sparse attention parameters** | | | | | | |
| Block attention size: 1024 | | | | | | |
| Sparse block size: 128 | | | | | | |
| Sparse block count: 24 (stage 1), 12 (stage 2), 4 (bottleneck) | | | | | | |

**Input embedding.** Each spatial location on the latitude-longitude grid constitutes one token. For a history of $T=4$ consecutive timesteps, the $C_{\text{dyn}}=82$ dynamic channels at each grid point are concatenated along the channel dimension. This vector is further concatenated with the static variables augmented with Cartesian unit-sphere coordinates ($C_{\text{static}}=6$) and sinusoidal time embeddings ($C_{\text{time}}=4$), yielding a total input dimension of $T \times C_{\text{dyn}} + C_{\text{static}} + C_{\text{time}} = 338$ per token. A preprocess MLP (Linear $\rightarrow$ RMSNorm $\rightarrow$ SiLU $\rightarrow$ Linear $\rightarrow$ RMSNorm) projects this to the hidden dimension of Stage 1, after which features are interpolated to the HEALPix mesh.

**HEALPix mesh.** The HEALPix grid is generated with the `healpy` library using *nested* pixel ordering, which groups spatially nearby pixels into contiguous memory locations. The resolution parameter $N_{\text{side}}$ determines the total number of pixels as $12N_{\text{side}}^2$: Stage 1 operates at $N_{\text{side}} = 64$

(49,152 pixels), Stage 2 at $N_{\text{side}} = 32$ (12,288 pixels), and the bottleneck at $N_{\text{side}} = 16$ (3,072 pixels).

**Downsampling and upsampling.** Transitions between resolution levels follow the HEALPix quad-tree hierarchy with a downsampling factor of $f=4$. During downsampling (Section 4), $W_x^{\downarrow} \in \mathbb{R}^{d_{\text{out}} \times 4d_{\text{in}}}$ projects the stacked child features and $W_p^{\downarrow} \in \mathbb{R}^{d_{\text{out}} \times 12}$ provides a position-aware bias from the $4 \times 3$ relative Cartesian coordinates, followed by RMSNorm. During upsampling, $W_x^{\uparrow} \in \mathbb{R}^{4d_{\text{out}} \times d_{\text{in}}}$ expands each coarse pixel to four fine pixels and $W_p^{\uparrow} \in \mathbb{R}^{4d_{\text{out}} \times 12}$ again provides a position-aware bias. The encoder skip connection is added element-wise after the upsampling projection, followed by RMSNorm.

**Output head.** After the final decoder stage, features are normalized via RMSNorm, interpolated from HEALPix back to the latitude-longitude grid (Eq. 4 with source and target reversed), and passed through a postprocess MLP (RMSNorm → Linear → SiLU → Linear). The final linear layer maps to $C_{\text{dyn}} = 82$ output channels with no activation function. The model predicts a normalized next state directly; denormalization recovers physical units at inference time.

**Cross-attention interpolation.** In the interpolation module (Eq. 4), queries are derived from geometry with $W_q \in \mathbb{R}^{d \times 3}$ acting on $L_2$-normalized relative Cartesian positions, while keys and values are derived from RMSNorm-normalized source features via $W_k, W_v \in \mathbb{R}^{d \times d}$. An output projection $W_o \in \mathbb{R}^{d \times d}$ is applied after the attention-weighted sum. All projections are fully learned; neighbor indices and relative positions are fixed after initialization.

**Noise injection details.** The noise vector $\mathbf{z} \in \mathbb{R}^{32}$ is sampled once per forward pass from $\mathcal{N}(\mathbf{0}, \mathbf{I})$ and transformed by a learned layer $W_z \in \mathbb{R}^{32 \times 32}$. In each cSwiGLU block (Section 4), a per-layer projection $W_n \in \mathbb{R}^{d_{\text{ff}} \times 32}$ maps $\mathbf{z}$ to the feed-forward hidden dimension. Both $W_z$ and all $W_n$ are initialized with near-zero weights $\sim \mathcal{N}(0, 0.01)$, so that noise injection has negligible effect at the start of training and its influence is learned gradually. The same $\mathbf{z}$ is broadcast to every transformer block across the encoder, bottleneck, and decoder, acting as a global latent variable. Different ensemble members receive independently sampled noise vectors.

## A.4 TRAINING SCHEDULE

Table 6 shows the complete training schedule. Pretraining on ERA5 (1979-2021) is followed by finetuning on HRES-fc0 (2016-2021) with progressively longer autoregressive rollouts.

Table 6: Training schedule details.

| Stage | Steps | AR Length | Learning Rate | Weight Decay | Schedule |
|---|---|---|---|---|---|
| Pretrain | 250k | 1 | 1e-3 | 1e-2 | Cosine (1e-3 → 1e-6), no warmup |
| Finetune 1 | 30k | 1 | 1e-4 | 1e-2 | Cosine decay by 1e-2 |
| Finetune 2 | 10k | 2 | 1e-5 | 1e-2 | Cosine decay by 1e-2 |
| Finetune 4 | 5k | 4 | 5e-6 | 1e-2 | Cosine decay by 1e-2 |
| Finetune 8 | 2.5k | 8 | 1e-6 | 1e-2 | Cosine decay by 1e-2 |
| Finetune 12 | 2.5k | 12 | 1e-6 | 1e-2 | Cosine decay by 1e-2 |
| **Common parameters** | | | | | |
| Batch size: 2, Ensemble size during training: 2 | | | | | |
| Optimizer: Muon (Jordan et al., 2024), Precision: float16 | | | | | |

**Optimizer.** We use Muon (Jordan et al., 2024) with momentum $\beta = 0.95$, Nesterov acceleration enabled, 5 Newton-Schulz orthogonalization steps, and no weight decay beyond what is specified per stage in Table 6. The base learning rate is $0.02$; per-stage values are listed in Table 6 and follow the cosine schedule described therein.

**Learning rate warmup.** Pretraining uses no warmup. All finetuning stages employ a 500-step linear warmup from $10^{-6} \times \eta$ to the stage-specific learning rate $\eta$, followed by cosine annealing.

**Early stopping.** We apply early stopping based on validation loss at each stage. For pretraining, the criterion is the single-step (6 h) prediction loss. For finetuning stage $k$ (with $k$ autoregressive rollout steps), the criterion is the loss on the $k$-th predicted step. The best checkpoint from each stage initializes the next.

**Gradient clipping.** We clip gradient norms to a maximum of 1.0 at every training step across all stages.

**Distributed training.** All experiments use distributed data parallelism (DDP) across 8 NVIDIA H100 GPUs with a per-GPU batch size of 2 (effective batch size 16). Training is performed entirely in float16 precision.

**Weight initialization.** All linear layers are initialized with $\mathcal{N}(0, \sigma)$ where $\sigma = \frac{1}{\sqrt{d_{\text{in}}}} \min(1, \sqrt{d_{\text{out}}/d_{\text{in}}})$, and all biases are set to zero. Residual-path layers (the gate of SwiGLU ($W_{13b}$), the attention output projection ($W_o$), noise bias projections ($W_n$), noise generator ($W_z$), and upsampling projections) are initialized with $\mathcal{N}(0, 0.01)$ so that residual contributions are near-identity at the start of training.

**Autoregressive rollout during finetuning.** Finetuning stages with $k > 1$ autoregressive steps use the pushforward trick: the first $k-1$ rollout steps are computed without gradients (i.e., with stopped gradients), and only the final $k$-th prediction step receives gradients. This avoids backpropagation through the full rollout chain, keeping memory requirements constant regardless of the number of rollout steps.

**Training ensemble generation.** During training, we use an ensemble of size $N=2$. At the first autoregressive step, the input state is replicated $N$ times within a single forward pass, and each replica receives an independently sampled noise vector $\mathbf{z}$. For subsequent rollout steps, each member evolves independently with a single noise sample ($N=1$). The CRPS loss (Section A.5) is computed over the resulting $N$-member ensemble. At inference, the ensemble size is increased to 48.

### A.5 Loss Function

The training objective is the latitude-weighted, variable-weighted fair CRPS (Eq. 10):

$$\mathcal{L} = \frac{1}{|D|} \sum_{d \in D} \frac{1}{HW} \sum_{h,w} \sum_{i=1}^{C} \alpha_i \, \omega_h \, \text{CRPS}(\hat{x}_{i,h,w,d}^{1:N}, \, \hat{y}_{i,h,w,d}), \tag{11}$$

where $d$ indexes the batch, $(h, w)$ indexes spatial grid points on the $H \times W$ latitude-longitude grid, $i$ indexes the $C=82$ output channels, $\alpha_i$ is the per-channel variable weight, and $\omega_h$ is the latitude weight. Both predictions $\hat{x}$ and targets $\hat{y}$ are in standardized (zero-mean, unit-variance) space.

**Variable-level loss weights.** Following Lam et al. (2023), pressure-level variables are weighted proportionally to their pressure level $p$ (in hPa), normalized by the mean pressure across all 13 levels ($\bar{p} = 463.46$ hPa): $\alpha(p) = p/\bar{p}$. This assigns higher weight to lower-tropospheric levels, reflecting both their greater meteorological importance and the larger magnitude of atmospheric variability near the surface. Surface variables receive fixed weights. Table 7 lists all weights.

**Latitude weighting.** To account for the convergence of meridians toward the poles, each grid row at latitude $\phi_h$ is weighted proportionally to the area it represents:

$$\omega_h = \frac{\cos(\phi_h)}{\frac{1}{H} \sum_{h'=1}^{H} \cos(\phi_{h'})}, \tag{12}$$

so that the weights average to unity across latitudes.

**Loss normalization.** The loss is computed in standardized space: both model outputs and targets are normalized per channel using the training-set mean and standard deviation (Section A.2). The

Table 7: Variable-level loss weights $\alpha_i$. All six pressure-level variables share the same per-level weight.

| Variable / Level | $\alpha_i$ |
|---|---|
| *Surface variables* | |
| 2-meter temperature | 1.0 |
| 10-meter U-wind | 0.1 |
| 10-meter V-wind | 0.1 |
| Mean sea level pressure | 0.1 |
| *Pressure levels (shared across all 6 pressure-level variables)* | |
| 1000 hPa | 2.157 |
| 925 hPa | 1.996 |
| 850 hPa | 1.834 |
| 700 hPa | 1.510 |
| 600 hPa | 1.294 |
| 500 hPa | 1.079 |
| 400 hPa | 0.863 |
| 300 hPa | 0.647 |
| 250 hPa | 0.539 |
| 200 hPa | 0.431 |
| 150 hPa | 0.324 |
| 100 hPa | 0.216 |
| 50 hPa | 0.108 |

mean in the loss is taken over all dimensions (batch, latitude, longitude, and channels), with $\alpha_i$ and $\omega_h$ acting as importance multipliers. Training uses mixed-precision (float16) with gradient scaling via PyTorch's `GradScaler`. No additional loss normalization or scaling is applied.

### A.6 BLOCK-SPARSE ATTENTION IMPLEMENTATION

**Kernel Design**  We implement block-sparse attention in Triton, following the memory-efficient approach of FlashAttention Dao et al. (2022). The forward pass loads query blocks into SRAM and streams selected key-value blocks through, computing attention without materializing the full attention matrix. The backward pass computes gradients for keys and values by iterating over all query blocks and only loading those into memory that are connected to a given key-value pair. The original NSA implementation batches query heads sharing the same key-value head, requiring sufficiently large batch sizes[2] for the dot product operations to be executed efficiently on Tensor Cores. In BSA, operating at the block level relaxes this constraint: batching occurs naturally across queries within a block, which at sizes $\geq$128 satisfies tile requirements and enables arbitrary GQA group size.

**Computational cost analysis.**  Let $b$ denote the block size yielding $\frac{N}{b}$ blocks in total. The combined cost of block-sparse attention across branches is: $\mathcal{O}\big(N^2/b^2 + Nnb + Nb\big)$. The compression branch (CG) computes all pairwise interactions between coarse-grained blocks resulting in $\mathcal{O}(N^2/b^2)$ cost. For block sizes $b \geq 128$, this cost does not form a bottleneck. The selection branch (FG) computes, for each token, attention over $n$ selected blocks of size $b$ yielding $\mathcal{O}(Nnb)$ cost, which is linear in $N$ as $n$ and $b$ are fixed hyperparameters. Local attention is $\mathcal{O}(Nb)$ since each token only attends to tokens within the same block. Together, these costs make block-sparse attention scalable to weather grids with hundreds of thousands of tokens. In practice, our BSA achieves a 2.5-9.4$\times$ wall-clock speedup over NSA, with the gap widening at longer sequences, see Fig. 6. We implement BSA in Triton Tillet et al. (2019) following the memory-efficient approach of FlashAttention Dao et al. (2022); see Appendix A.3 for details.

**Block-sparse attention vs. native sparse attention.**  Figure 6 compares the runtime of our block-sparse attention (BSA) against native sparse attention (NSA) (Yuan et al., 2025). BSA achieves 2 to

---

[2]At least 16 along each dimension in Triton.

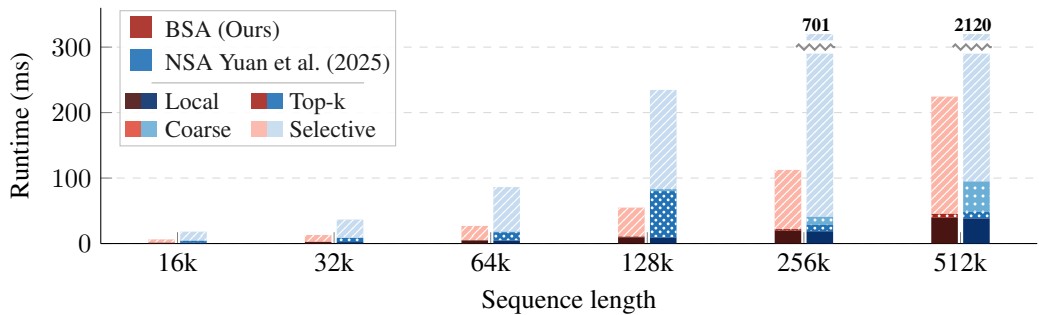

Figure 6: Runtime comparison of native sparse attention (NSA) vs block-sparse attention (BSA, ours) on NVIDIA A4500. NSA implementation from Yang & Zhang (2024). BSA achieves consistent speedups across all sequence lengths.

$3\times$ faster execution across sequence lengths relevant to weather forecasting. This advantage stems from BSA's block-structured memory access patterns, which better utilize GPU tensor cores compared to NSA's irregular sparsity patterns. The efficiency gap widens at longer sequences, making BSA particularly suitable for high-resolution atmospheric grids.

### A.7 COMPUTATIONAL EFFICIENCY

**Training efficiency.** As detailed in Table 2, MOSAIC requires only 16 GPU-days total, operating entirely in float16 precision. In contrast, GraphCast trains for 4 weeks on 32 TPUv4s, GenCast for 5 days on 32 TPUv5s, and FGN accumulates 490 TPU-days of compute, all using float32 precision. The combination of Muon optimizer, float16 precision, and block-sparse attention enables MOSAIC to achieve competitive performance with approximately $30\times$ less compute than comparable probabilistic models.

**Inference speed.** MOSAIC generates a 24-member ensemble forecasting 12 days ahead in approximately 1 minute on a single H100 GPU at $1.5°$ resolution.

### A.8 EVALUATION PROTOCOL

**Inference ensemble generation.** At inference, we generate a 48-member ensemble following the same branching scheme as during training (Section A.4): all members share the same initial condition and diverge at the first autoregressive step through independently sampled noise vectors. Each member then evolves autoregressively with a single noise sample per step.

**Baseline results.** All baseline scores reported in this paper are obtained from the WeatherBench 2 public evaluation framework (Rasp et al., 2023). Before computing metrics, WeatherBench 2 first-order conservatively regrids all forecasts and ground truths to $1.5°$ resolution, ensuring a common evaluation grid. Since MOSAIC operates natively at $1.5°$, no additional regridding is required for our model. We do not re-run baseline models; we use their publicly available forecast outputs evaluated through WeatherBench 2.

### A.9 REPRODUCIBILITY

**Software stack.** All experiments are implemented in PyTorch 2.8 with CUDA 12.8. Custom block-sparse attention kernels are written in Triton 3.6. Key dependencies include `einops` (tensor rearrangement), `healpy` (HEALPix grid generation), `scikit-learn` (BallTree for neighbor lookup), `xarray` and `zarr` (data loading from WeatherBench 2), and `wandb` (experiment tracking). Spectral analysis uses `pyshtools` and `scipy`.

**Code and data availability.** Code will be released upon publication. Training and evaluation data are publicly available through the WeatherBench 2 repository (Rasp et al., 2023) on Google Cloud Storage.

| | RES. | STEP | V850 | U10M | V10M |
|---|---|---|---|---|---|
| IFS HRES | 0.1° | 1H | 1.206 | 0.833 | 0.872 |
| SPHERICALCNN | 1.4° | 6H | 1.471 | N/A | N/A |
| STORMER | 1.4° | 24H | 1.169 | 0.759 | 0.781 |
| NEURALGCM ENS (50) | 1.4° | 12H | 1.256 | N/A | N/A |
| ARCHESWEATHER-MX4 | 1.5° | 24H | 1.204 | 0.750 | 0.781 |
| ARCHESWEATHERGEN (MEAN) | 1.5° | 24H | 1.190 | **0.743** | **0.775** |
| MOSAIC (MEAN; OURS) | 1.5° | 6H | 1.146 | 0.760 | 0.790 |
| MOSAIC$_\triangle$ (MEAN; OURS) | 1.5° | 6H | **1.144** | 0.767 | 0.806 |

Table 8: Comparison of coarse resolution MLWPs on RMSE scores for key weather variables with 24h lead-time against ERA5 data, 2020 test year (continuation of Table 1). Best scores in **bold**, second best underlined.

| | RES. | Z500 | T850 | Q700 | U850 | V850 | T2M | SP | U10M | V10M |
|---|---|---|---|---|---|---|---|---|---|---|
| IFS HRES | 0.1° | 41.23 | 0.634 | 537.3 | 1.126 | 1.149 | 0.541 | 58.70 | 0.802 | 0.832 |
| IFS ENS | 0.2° | 40.91 | 0.630 | 503.1 | 1.102 | 1.121 | 0.589 | 59.44 | 0.784 | 0.806 |
| PANGU | 0.25° | 45.01 | 0.681 | 537.2 | 1.180 | 1.218 | 0.624 | 59.54 | 0.796 | 0.825 |
| GRAPHCAST | 0.25° | 38.32 | 0.570 | 475.3 | 1.021 | 1.048 | 0.476 | 49.36 | 0.692 | 0.719 |
| GENCAST | 0.25° | 39.20 | 0.570 | 480.7 | 1.040 | 1.070 | 0.462 | 50.04 | 0.701 | 0.732 |
| FGN | 0.25° | 32.27 | 0.510 | 443.2 | 0.928 | 0.953 | 0.418 | 42.22 | 0.627 | 0.655 |
| AURORA | 0.25° | 38.12 | 0.566 | 477.9 | 1.003 | 1.031 | 0.474 | 48.10 | 0.681 | 0.710 |
| MOSAIC (OURS) | 1.5° | 36.78 | 0.572 | 482.8 | 1.028 | 1.054 | 0.525 | 47.91 | 0.701 | 0.729 |

Table 9: RMSE scores for key weather variables at 24 h lead time against HRES-f. All metrics are evaluated at 1.5° resolution following the WeatherBench 2 protocol (Rasp et al., 2023).

## B  ADDITIONAL RESULTS

### B.1  ADDITIONAL FORECAST SKILL RESULTS

Figures 7–9 extend the main-text evaluation (Fig. 4) to the remaining surface and pressure-level variables: 10-meter V-wind, mean sea level pressure, U- and V-wind at 850 hPa, and specific humidity at 700 hPa.

|  | RES. | Z500 | T850 | Q700 | U850 | V850 | T2M | SP | U10M | V10M |
|---|---|---|---|---|---|---|---|---|---|---|
| IFS HRES | 0.1° | 809.0 | 3.626 | 1847. | 6.417 | 6.500 | 2.576 | 753.3 | 4.513 | 4.767 |
| IFS ENS | 0.2° | 621.9 | 2.761 | 1390. | 4.854 | 4.918 | 1.991 | 574.7 | 3.394 | 3.595 |
| PANGU | 0.25° | 799.8 | 3.573 | 1782. | 6.267 | 6.334 | 2.648 | 742.4 | 4.378 | 4.633 |
| GRAPHCAST | 0.25° | 747.6 | 3.364 | 1601. | 5.953 | 6.032 | 2.415 | 698.9 | 4.195 | 4.436 |
| GENCAST | 0.25° | 607.5 | 2.693 | 1340. | 4.758 | 4.826 | 1.932 | 563.2 | 3.325 | 3.521 |
| FGN | 0.25° | 583.1 | 2.595 | 1302. | 4.636 | 4.710 | 1.864 | 542.7 | 3.247 | 3.440 |
| AURORA | 0.25° | 668.4 | 2.979 | 1435. | 5.060 | 5.051 | 2.200 | 624.9 | 3.542 | 3.718 |
| MOSAIC (OURS) | 1.5° | 610.4 | 2.703 | 1359. | 4.773 | 4.828 | 1.992 | 563.2 | 3.339 | 3.534 |

Table 10: RMSE scores for key weather variables at 240 h (10-day) lead time against HRES-fc0. All metrics are evaluated at 1.5° resolution following the WeatherBench 2 protocol (Rasp et al., 2023).

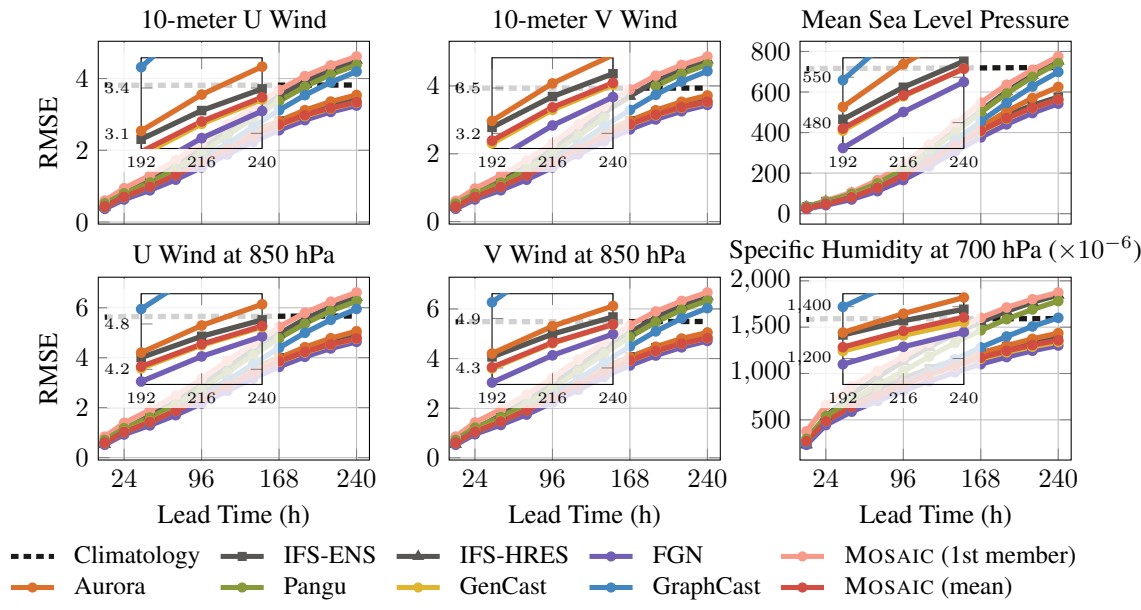

Figure 7: RMSE as a function of lead time for additional variables not shown in the main text. Top row: surface variables (10-meter U-wind, 10-meter V-wind, mean sea level pressure); bottom row: pressure-level variables (U850, V850, Q700). All forecasts are regridded to 1.5° resolution following the WeatherBench 2 protocol (Rasp et al., 2023).

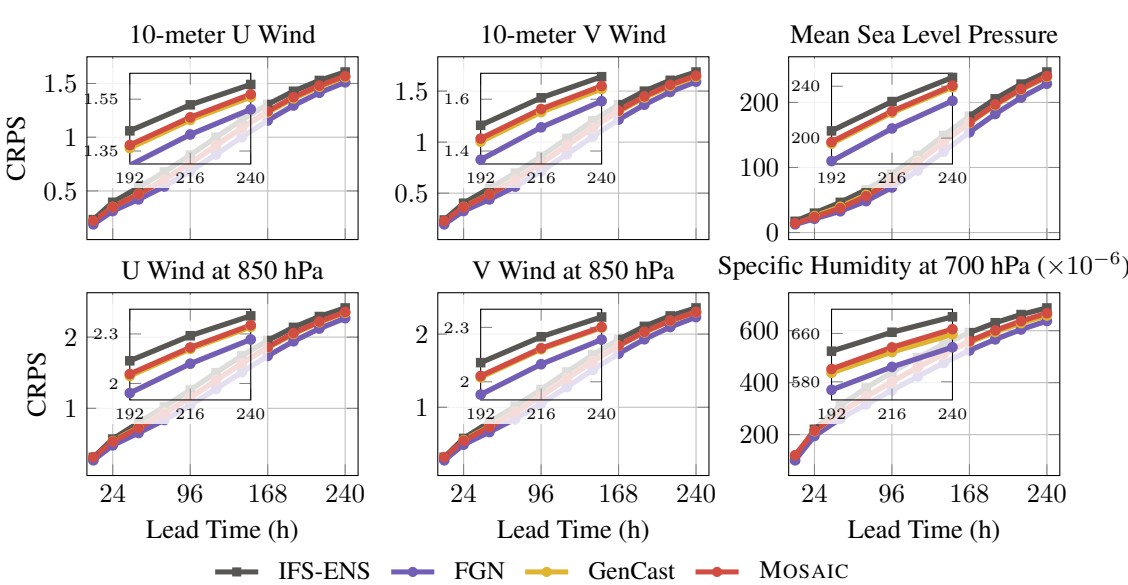

Figure 8: CRPS as a function of lead time for the same variables as Fig. 7. Top row: surface variables; bottom row: pressure-level variables. All forecasts are regridded to 1.5° resolution following the WeatherBench 2 protocol (Rasp et al., 2023).

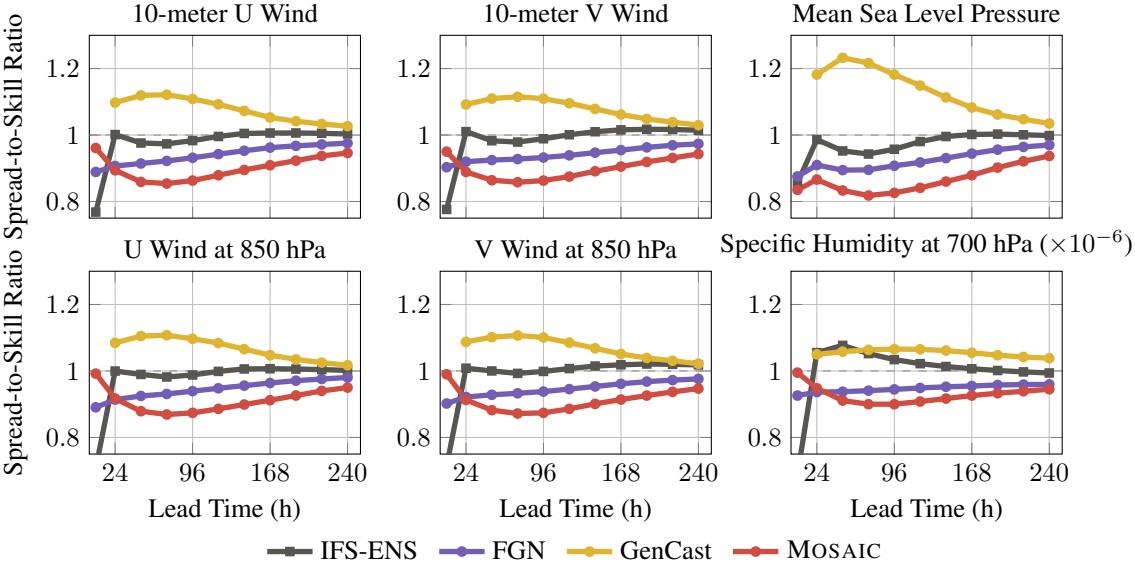

Figure 9: Spread-to-skill ratio as a function of lead time for the same variables as Fig. 7. Values close to 1.0 (dashed line) indicate well-calibrated ensembles. Top row: surface variables; bottom row: pressure-level variables.

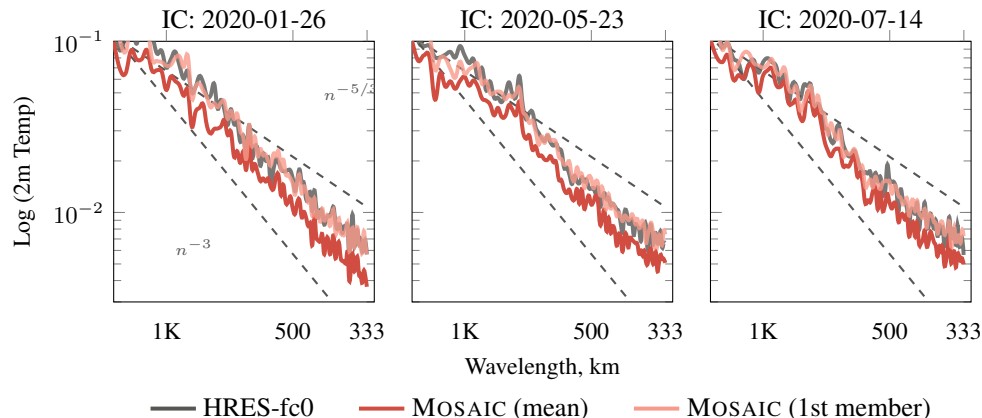

Figure 10: Examples of global temperature spectra at 2-meter height for 10-day MOSAIC forecasts compared to HRES-fc0 0.25° ground truth, shown for multiple initial conditions (all start at 00:00 UTC).

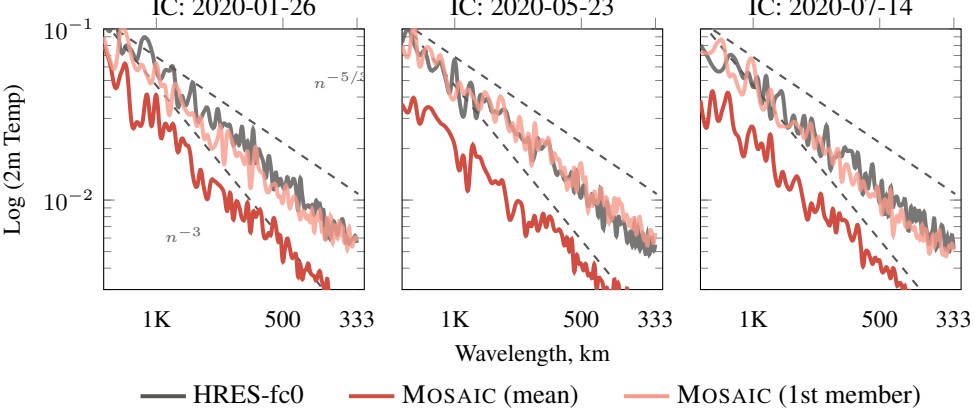

Figure 11: Examples of global kinetic energy spectra at 10-meter height for 10-day MOSAIC forecasts compared to HRES-fc0 0.25° ground truth, shown for multiple initial conditions (all start at 00:00 UTC).

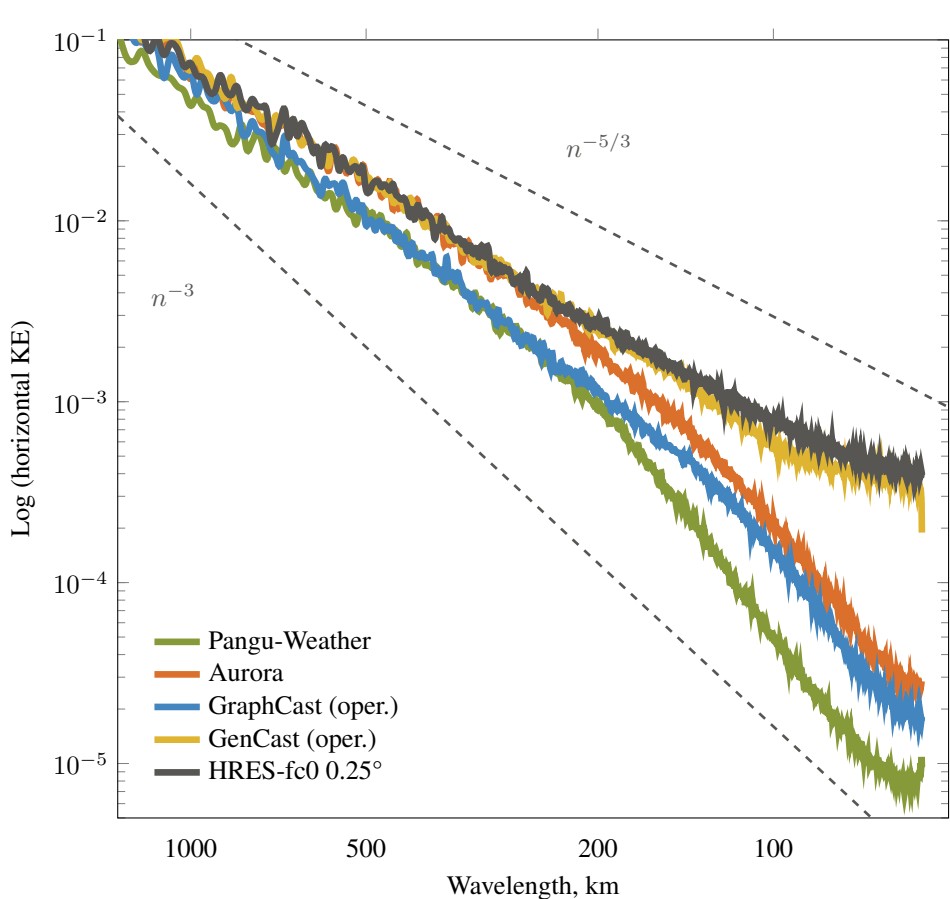

Figure 12: Global kinetic energy spectra at 10-meter height for 3-day forecasts from 0.25° state-of-the-art ML weather prediction models, compared to the IFS HRES reference. Divergence from the ground truth is most pronounced at fine spatial scales for deterministic models.

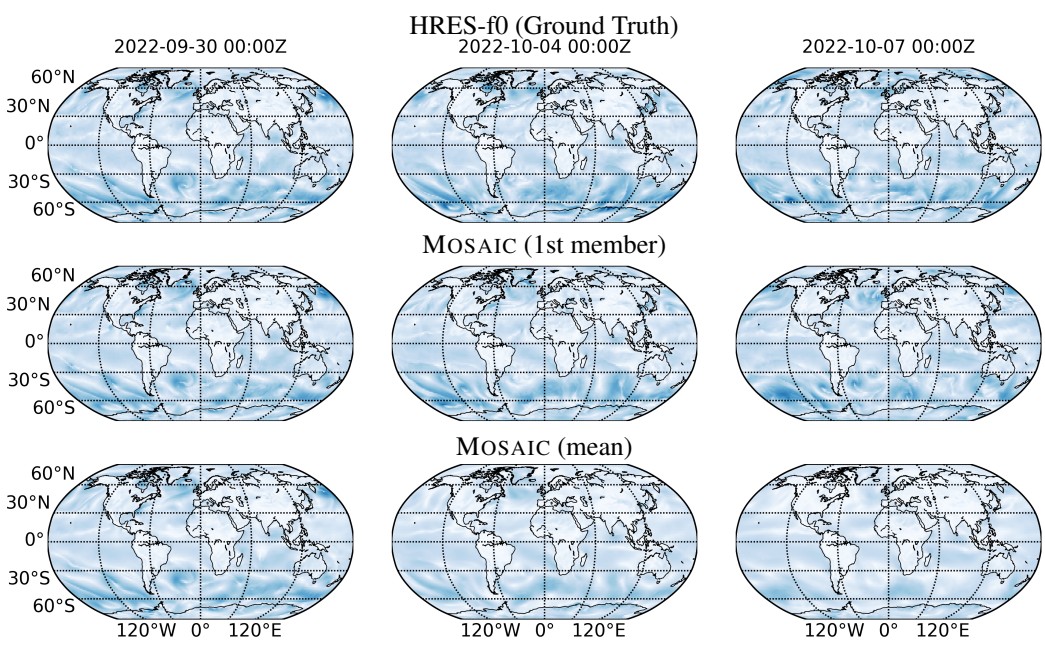

Figure 13: Forecast rollout trajectories showing 10-day evolution of wind speed fields at 850 hPa.

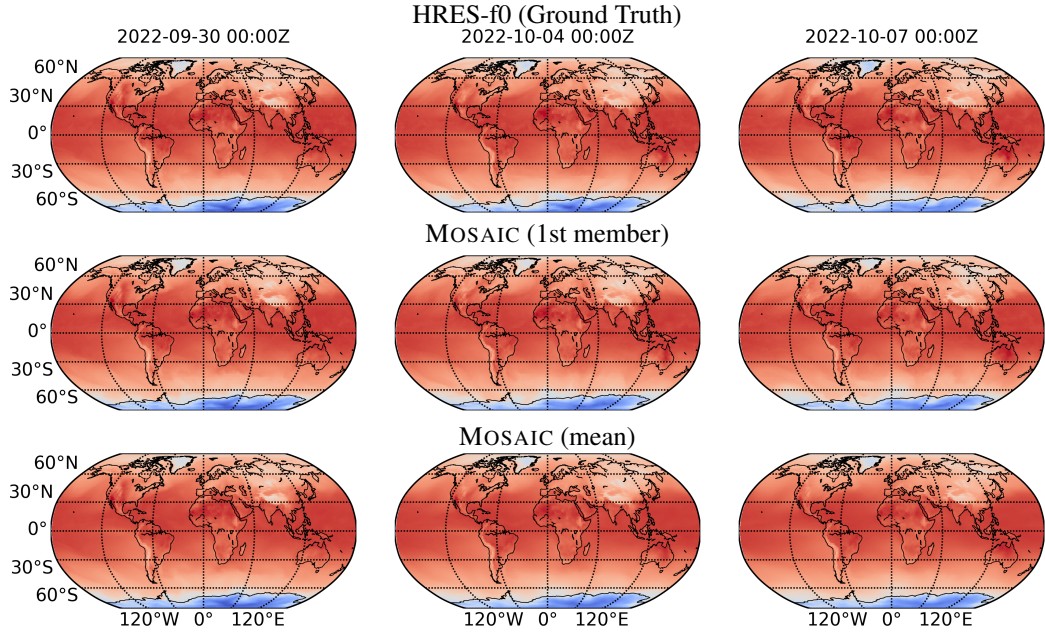

Figure 14: Forecast rollout trajectories showing 10-day evolution of surface temperature fields.

