# OpenReview forum: "(Sparse) Attention to the Details: Preserving Spectral Fidelity in ML-based Weather Forecasting Models"
_ICLR.cc/2026/Workshop/FM4Science — ICLR 2026 Workshop FM4Science Poster_

### Official Review · Reviewer_VLfH · 2026-02-23
**Preserving Spectral Fidelity in Machine Learning Weather Forecasting**

**Rating:** 7
**Confidence:** 4

**Review:**

The weather forecast is a scientific question, which has been modeled using ML-driven models in the past decade (MLWPs). The authors first pointed out the current limitations of WLWPs and focused on the modeling of high dynamics of realistic weather, aka spectral degradation. The authors proposed MOSAIC based on a light-weight probabilistic transformer-based weather model. The main contribution to weather forecasting from this paper includes using block-sparse attention to achieve the backbone attention mechanism while preserving efficiency and lower computational cost. In the context of learning, the authors demonstrated the way to directly optimize the spectral degradation by adding specific spectral losses. Although this paper did not achieve all SOTA results compared with previous models, this paper did provide promising training efficiency and lower computational resources, which potentially generate insights for future research directions.

Major concerns:
1. Since this framework prioritizes spectral fidelity over pure RMSE, the authors need to further explain the fact that this framework does not achieve SOTA results compared with previous models.
2. 1.5 resolution is kind of rough compared with SOTA models using 0.25. How to make sure node spectral degradation is not due to rough resolution?
3. No ablation was tested. I am especially curious about other resolution degrees and day length used in rollouts.

Minor concerns:
1. Sparse attention hyperparameters not justified.
2. FLOP needs to be reported.
3. Any failure cases can be discussed?

---

### Official Review · Reviewer_KyKy · 2026-02-23
**A Promising Sparse-Attention Framework for Probabilistic Weather Prediction With Open Evaluation Questions**

**Rating:** 7
**Confidence:** 4

**Review:**

**Summary**
This paper presents MOSAIC, a probabilistic transformer architecture for global weather forecasting that targets two fundamental shortcomings of current ML-based weather prediction models: spectral energy loss and over-smoothing due to compressive bottleneck. By combining a hardware-aware block-sparse attention mechanism aligned with HEALPix grids and learned functional perturbations for efficient ensemble generation. The approach is technically sound and system-level original, particularly in its adaptation of sparse attention to spherical scientific data.
Empirically, MOSAIC demonstrates strong spectral fidelity and competitive probabilistic performance; however, its deterministic accuracy (e.g., RMSE of individual ensemble members) is weaker than several baselines, and the claim regarding better spectral alignment would benefit from more direct comparisons and clearer attribution of gains between architectural and probabilistic components.

**Pros**
- **Clear motivation:** The framing of the motivation and the two identified failure modes: spectral energy loss and non-probabilistic forecasting, is accurate and well grounded in both the NWP and ML literature.
- **In-time new technique adaptation:** It is a clever adaptation of NSA to the scientific domain, especially through the use of HEALPix for coarse attention computation to enhance standard NSA with continuous, mesh-aligned heuristics. The authors also record inference-time speedups compared to NSA.
- **Efficient ensemble generation:** Uncertainty is introduced inside the network through learned perturbations, enabling low-cost ensemble sampling without diffusion-style generation while preserving fine-scale variability.
- **Comprehensive baseline comparison:** The method is evaluated against a broad set of competitive MLWP models, including both probabilistic and deterministic methods.
- **Nice visualizations:** Figures 1,2,3 are informative and aesthetically well designed. The color scheme is consistent and coherent  throughout the paper.

**Cons**
- **Weaker deterministic accuracy:** At least in Figure 4, the performance of MOSAIC (1st member) is reported to be worse than most baselines in terms of RMSE. The improvement is only from MOSAIC (mean). This raises a fairness concern when comparing deterministic baselines to the MOSAIC (mean), since the learned functional perturbations used for ensemble generation could also be incorporated into other deterministic baselines.
- **Limited spectral comparison:** Showing only the spectral degradation of mean predictions at high frequencies provides limited insight. Figure 12 presents baseline performance, yet it does not include MOSAIC, and its axis ranges differ from those used in the MOSAIC spectral figures, making it hard to compare directly.
- **Insufficient ablation of architectural vs stochastic effects:** The relative contributions of block-sparse attention and probabilistic training are not fully disentangled through ablation studies.
- **Main-text emphasis skewed toward method:** The main text focus too much on the architectural details, while some key empirical findings are deferred to the appendix.
- **Restricted evaluation regime:** Experiments are limited to a single (1.5°) resolution, leaving scalability to higher resolutions or other regimes untested.

---

### Meta-Review · Area_Chair_sKS7 · 2026-02-27

**Recommendation:** Accept (Oral)
**Confidence:** 5

**Metareview:**

Strong accept.

---

### Decision · Program_Chairs · 2026-03-03

Accept (Oral)